# TAEGAN: Generating Synthetic Tabular Data for Data Augmentation

## Abstract

Synthetic tabular data generation has gained significant attention for its potential in data augmentation, software testing and privacy-preserving data sharing. However, most research has primarily focused on larger datasets and evaluating their quality in terms of metrics like column-wise statistical distributions and inter-feature correlations, while often overlooking its utility for data augmentation, particularly for datasets whose data is scarce. In this paper, we propose Tabular Auto-Encoder Generative Adversarial Network (TAEGAN), an improved GAN-based framework for generating high-quality tabular data. Although large language models (LLMs)-based methods represent the state-of-the-art in synthetic tabular data generation, they are often overkill for small datasets due to their extensive size and complexity. TAEGAN employs a masked auto-encoder as the generator, which for the first time introduces the power of self-supervised pre-training in tabular data generation so that essentially exposes the networks to more information. We extensively evaluate TAEGAN against five state-of-the-art synthetic tabular data generation algorithms. Results from 10 datasets show that TAEGAN outperforms existing deep-learning-based tabular data generation models on 9 out of 10 datasets on the machine learning efficacy and achieves superior data augmentation performance on 7 out of 8 smaller datasets. Code is available at: https://anonymous.4open.science/r/taegan-2AB4

## 1 Introduction

The incentive for generating synthetic tabular data has been mostly data augmentation, software testing, and privacy-preserving data sharing. However, recent research in this field has been focused on the latter two use cases, which has a wider application on larger datasets. In this paper, we focus on data augmentation, which is particularly important when data is not abundant. Instead of only evaluating metrics like column-wise statistical distributions and inter-feature correlations, we assess the synthetic data generator's actual effectiveness in augmenting data. The current state-of-the-art synthetic tabular data generator has been based on large language models (LLMs) (Borisov et al., 2023; Solatorio & Dupriez, 2023;

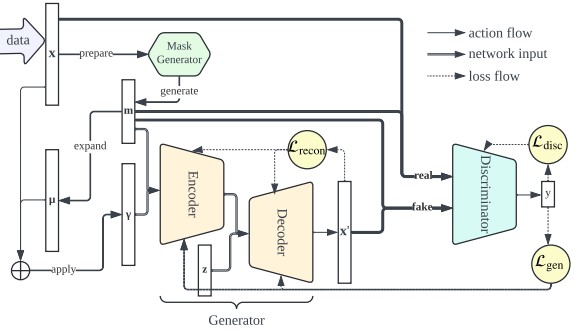

Figure 1: Architecture of TAEGAN network. The generator consists of an encoder and decoder output, and noise from latent space is inputted to decoder only. For all networks, if the arrows meet before directing to the network, it means they are concatenated before passed.

Gulati & Roysdon, 2023; Zhao et al., 2023). Although LLM-based models are expected to produce data with very high fidelity, but the way it is trained, which is by naïve next-token-predictions, makes it hard to extend the generator's ability beyond the training set. In particular, models with tens of millions of parameters would be a significant over-skill for small tables. Thus, these models may not achieve very good data augmentation effect because they cannot generalize more than what it has seen. In comparison, previous state-of-the-art models in the field, generative adversarial net-

works (GANs) (Goodfellow et al., 2014), can excel in generalization, provided that mode collapse is effectively controlled. This is because the core generator is trained indirectly through feedback from the discriminator, rather than being directly trained on real data.

In this paper, we propose a novel GAN architecture for tabular model generation: Tabular Auto-Encoder Generative Adversarial Network (TAEGAN). TAEGAN outperforms all existing deep learning models[1], including all GANs, on classical synthetic data metrics. It also achieves the best data augmentation effect among all models, including not only deep learning ones, but also LLM-based and tree-based ones. TAEGAN essentially changes the generator of GAN into a masked auto-encoder (see Figure 1). By incorporating flexible masks and masked data as conditions, we eliminate the need for a classifier network, which is typically required in GANs for tabular data with a target column (Park et al., 2018; Zhao et al., 2021; 2024). This approach allows the generator to learn better inter-feature relationships. TAEGAN generates data with the highest quality among all deep-learning methods in 9 out of 10 test datasets, and best augments the downstream classification tasks in 7 out of 8 smaller datasets.

Main contributions of the paper include:

- A novel GAN architecture for tabular data that makes the generator a masked auto-encoder, so that the generator can be trained on a well-defined self-supervised pre-training task independently.
- An enhanced multivariate sampling method on tabular data to address data skewness in all features instead of only one at each time.
- A mixture of continuous and discrete noise construction for GAN's input for better capturing the clustered nature of data.
- An interaction loss introduced to enforce better inter-feature relation understanding, especially on highly-correlated columns.
- Performed an experimental comparison with five other synthetic tabular data generation models, empirically demonstrating TAEGAN's superiority in tabular data augmentation for small datasets.
- Conducted a theoretical analysis of the advantages, disadvantages, and potential failure modes of state-of-the-art models (see Appendix A).
- Proposed a novel approach for score normalization and the aggregation of multiple metrics to provide a more comprehensive and concise interpretation of data quality (see Appendix B.3).

## 2 RELATED WORK

### 2.1 GENERATING SYNTHETIC TABULAR DATA

The early history of synthetic tabular data generation has been mostly on statistical, or probabilistic approaches, such as decision trees (Reiter, 2005) and Bayesian networks (Zhang et al., 2017; Aviñó et al., 2018). Since deep neural network generative models, including GAN (Goodfellow et al., 2014), Variational auto-encoder (VAE) (Kingma & Welling, 2013), Diffusion model (Ho et al., 2020), and transformers (Vaswani et al., 2017), start to gain their popularity, they have also dominated the research on tabular data generation, although there are still new statistical models coming out (Watson et al., 2023). In particular, before Diffusion model and transformers started to show their powerfulness in complex image and text tasks, GANs have dominated in tabular data generation (Park et al., 2018; Xu et al., 2019; Zhao et al., 2021; 2024). Later, transformers were proven especially powerful, especially as LLM, the dominating model for tabular data also shifted to transformers. Borisov et al. (2023) first introduced transformers into synthetic tabular data generation, and following that, a few improved transformers for tabular data were proposed (Solatorio & Dupriez, 2023; Zhao et al., 2023; Gulati & Roysdon, 2023), and there are works finetuning foundational LLMs for tabular data (Ling et al., 2024; Seedat et al., 2024). Yet indeed, there are also tabular generative models using VAE (Xu et al., 2019) and Diffusion model (Kotelnikov et al., 2023).

In this paper, we leave out the foundational LLMs (other LLM-based methods are still considered). While we do believe that they are promising in data augmentation by introducing world knowledge, they are so far more designed for low-data to no-data scenarios, where the number of data rows are only sufficient to demonstrate the data format, rather than giving sufficient information for actual

---

[1]In this paper, we refer to non-LLM neural networks as deep learning methods.

data distribution and correlation. Also, these methods usually need well-documented description of the datasets, which is not always available in real life. Therefore, we stick to the methods that only look at the current dataset.

## 2.2 TABULAR DATA AUGMENTATION

While data augmentation is one of the main objectives for synthetic tabular data generation, it has long been a standalone research topic for decades. Besides directly generating synthetic data out of generators (Manousakas & Aydöre, 2023), there are a variety of different methods for tabular data augmentation. They include methods mainly for data rebalancing (Chawla et al., 2002; Batista et al., 2004; He et al., 2008), augmentation policy learning (Cubuk et al., 2018; Zhou et al., 2021), and data interpolation and noise injection (Zhang et al., 2018; Lim et al., 2022).

In this paper, we will focus on the usefulness of synthetic data from generation models in data augmentation without elaborate analysis of other methods, as synthetic data can be applied in parallel with or on top of many of the other augmentation techniques (da Silva et al., 2021; Qian et al., 2023).

## 2.3 USE OF MASKS IN TRAINING AND PRE-TRAINING

Masking is an important technique for self-supervised learning. Masked language modeling (MLM) has been a primary task for text pre-training (Devlin et al., 2019; Liu et al., 2019), and masked image modeling (MIM) has played a similar role in image pre-training (He et al., 2022; Bao et al., 2022). Data generation using masked modeling by gradually replacing masks with generated data has also gained great success in images (Chang et al., 2022).

In tabular data, masking can be applied for imputation (Du et al., 2024), and recent work has also explored its use for tabular data generation. REaLTabFormer (Solatorio & Dupriez, 2023) introduces *target masking*, a technique that randomly replaces label tokens with a special mask token to prevent overfitting. However, this approach is not strictly a form of masked modeling. TabMT (Gulati & Roysdon, 2023) is a true usage of masked modeling for generating tabular data, that combines masking with transformer for tabular data generation, but it suffers from long training and sampling problem due to its large model size and autoregressive sampling nature.

## 3 TAEGAN EXPLAINED

### 3.1 PRELIMINARIES AND DATA PROCESSING

TAEGAN inherits the data preprocessing of CTAB-GAN+ (Zhao et al., 2024), the state-of-the-art tabular GAN model. Categorical columns are one-hot encoded, while numeric columns undergo a series of transformations. First, if a numeric column has a long-tail distribution, it is log-transformed. Next, the column is decomposed using Variational Gaussian Mixtures (VGM) to effectively capture multi-modal distributions. Finally, each column is represented by concatenating the one-hot encoded mode with the numeric value within that mode (see details on *mode-specific normalization* in Xu et al. (2019)). Thus, for a table $\mathcal{T}$ with $M$ rows and $N$ columns, each column can be represented as a vector $\mathbf{x}_n, n \in \{1, 2, \ldots, N\}$. $\mathbf{x}_n$ consists of a discrete part $\mathbf{d}_n$ and continuous part $\mathbf{c}_n$, i.e. $\mathbf{x}_n = \mathbf{d}_n \oplus \mathbf{c}_n$, where $\oplus$ denotes vector concatenation. Note that $\mathbf{d}_n$ is always a one-hot vector. Let $|\cdot|$ be the number of dimensions of a vector, and let $D_n = |\mathbf{x}_n|, D_{dn} = |\mathbf{d}_n|, D_{cn} = |\mathbf{c}_n|$, so that $D_{dn}$ is the number of categories for categorical columns and number of modes for numeric columns, $D_{cn}$ is 0 for categorical columns and 1 for numeric columns, and $D_n = D_{dn} + D_{cn}$. Thus, a row in $\mathcal{T}$ can be represented by the vector $\mathbf{x} = \mathbf{x}_1 \oplus \mathbf{x}_2 \oplus \cdots \oplus \mathbf{x}_N$ with $D = \sum_{n=1}^{N} D_n$ dimensions.

To understand TAEGAN, it is important to understand the conditional vector and training-by-sampling mechanism proposed by Xu et al. (2019) for CTGAN, which is inherited by subsequent works (Zhao et al., 2021; 2024). The generator is conditional, that takes in not only noise, but also a conditional vector $\boldsymbol{\gamma}$ with $D_\gamma = \sum_{n=1}^{N} D_{dn}$ dimensions. The conditional vector is dependent on a mask indicator, represented as the binary vector $\mathbf{m} = \mathbf{m}_1 \oplus \mathbf{m}_2 \oplus \cdots \oplus \mathbf{m}_N$, where $|\mathbf{m}_n|$ is 1 if $n$-th column is categorical, and 2 if it is numeric. We call each individual dimension of $\mathbf{m}$ a component, so each column has either 1 or 2 components. Let $D_m = |\mathbf{m}|$ be the total number of components.

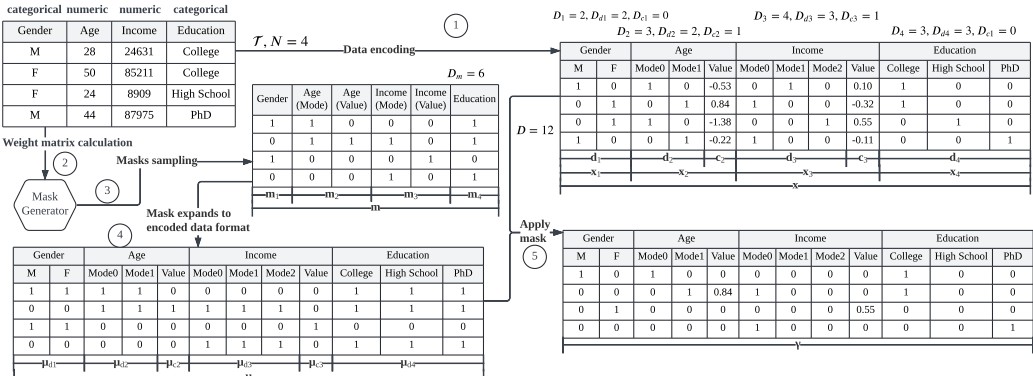

Figure 2: Example of TAEGAN data mask process. Prior works, including CTGAN (Xu et al., 2019) and CTAB-GAN+ Zhao et al. (2024) should conform to the constraint that only exactly one discrete component can be selected in the mask, namely, $\|\mathbf{m}\|_1 = 1$ and $\mathbf{m}_{ci} = 0, \forall i$, but the constraint is relaxed to any non-zero mask, namely, $\mathbf{m} \in \{0,1\}^{D_m} \setminus \{\mathbf{0}\}$ in TAEGAN (see Section 3.3.1).

The mask indicator can be expanded by the dimensions to create the data masks to be another binary vector $\boldsymbol{\mu} = \boldsymbol{\mu}_{d1} \oplus \boldsymbol{\mu}_{c1} \oplus \boldsymbol{\mu}_{d2} \oplus \boldsymbol{\mu}_{c2} \oplus \cdots \oplus \boldsymbol{\mu}_{dN} \oplus \boldsymbol{\mu}_{cN}$, where $|\boldsymbol{\mu}_{dn}| = D_{dn}, |\boldsymbol{\mu}_{cn}| = D_{cn}$. $\boldsymbol{\mu}_{dn}, \boldsymbol{\mu}_{cn}$ are always vectors with only one unique value for any $n$, and this value corresponds to the value in $\mathbf{m}_n$. For CTAB-GAN+ (Zhao et al., 2024), the conditional vector must satisfy the constraint that in $\mathbf{m}$, only one dimension can be 1 (the rest are 0, so $\|\mathbf{m}\|_1 = 1$), and it must correspond to a discrete component. Then, the mask $\mathbf{m}$, and hence $\boldsymbol{\mu}$, generates the conditional vector $\boldsymbol{\gamma} = \boldsymbol{\mu}^T \cdot \mathbf{x}$. Figure 2 shows an example with the notions mentioned previously summarized.

Then, in order to counter the imbalance of data, the training-by-sampling method is proposed such that during training, whenever a new sample is needed, the sample is selected by firstly randomly select $n \in \{1, 2, \ldots, N\}$, so that $\mathbf{m}_n$ is selected to have its categorical component as 1. Based on the probability mass function constructed by normalized logarithm-transformed smoothed-by-1 frequency of each value in this categorical component, randomly sample a value for this component. A row in $\mathcal{T}$ with this value in the selected component is thus sampled. During sampling, this same process is done but without logarithm-transformation and smoothing. Figure 3a shows an example.

## 3.2 Motivation and Structure

Zhao et al. (2021) confirms that an auxiliary classifier network, which is first introduced in Table-GAN (Park et al., 2018) and inherited in some other tabular GANs (Zhao et al., 2021; 2024), can help improve the performance of synthetic tabular data generation. However, not all tabular datasets include a designated target column for machine learning tasks, and the goal of synthetic data generation should extend beyond feature-to-target relationships to encompass all feature interactions. Therefore, instead of focusing only on a single target column, it is worth considering replacing the auxiliary classifier network with a more general prediction network that predicts all columns, including both features and the target. This makes an auto-encoder a suitable alternative, as it reconstructs the whole data. Nevertheless, an auto-encoder operates directly on raw input data, which may limit its ability to capture complex feature relationships. To address this, instead of feeding raw data as both the input and the expected output, we apply varying masks to the input data during training. This forces the auto-encoder to infer missing values based on the remaining features, thereby enhancing its ability to learn feature interdependencies. In this setup, the masked data, along with an indicator of which dimensions are masked, serves as the input, while the full raw data is the output. Reconstruction with masks has been shown to be an effective self-supervised pre-training task in other contexts (He et al., 2022), so we expect this approach to provide the generator with rich information about data distributions and relationships.

To maximize the capability of the auto-encoder to learn data relations, we can vary the masks not only in terms of the dimensions it masks out, but also the number of dimensions it masks out. The auto-encoder auxiliary network can thus be understood as a network with partial data known and full data to be recovered. This coincides somehow with the generator when the GAN is designed to

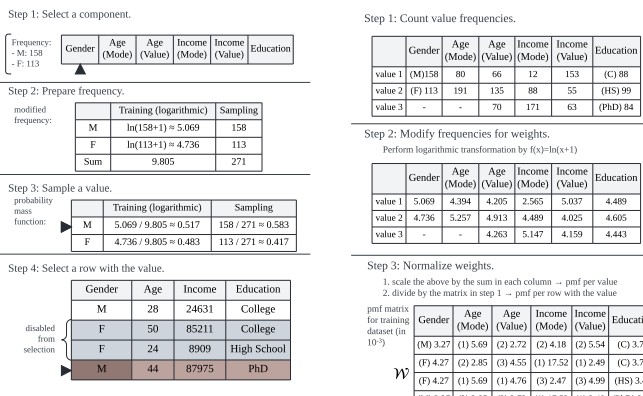

(a) Training-by-sampling from existing works (Xu et al., 2019) explained with the example.

(b) Weight matrix calculation explained with the example.

(c) Multivariate training-by-sampling using weight matrix explained with the example.

Figure 3: An illustration of the sampling process and weight matrix calculation. Black arrow indicates the selected one(s).

be conditional. Using a conditional generator for tabular data generation in GAN is first introduced in CTGAN (Xu et al., 2019) and has been empirically shown to be highly effective. This conditional generator takes in partially known data, which is one randomly selected categorical column, and generates the full raw data. The only difference between this conditional generator and the proposed auto-encoder is the number of dimensions masked. Therefore, it makes sense to combine the auto-encoder and conditional generator into the same network. The mask indicator and the masked data put together can be considered equivalent to the conditional vector of the generator of previous works. Noise is still necessary for the generator, but to ensure the encoder learns the tabular data without interference, the noise is concatenated to the encoder's output and used as input for the decoder. The discriminator can keep as it is. The resulting network structure can be found in Figure 1.

We base our work on the state-of-the-art GAN model on tabular data, CTAB-GAN+ (Zhao et al., 2024). We swapped the main architecture of CTAB-GAN+, which is CNN-based GAN, DC-GAN (Radford et al., 2016), to MLPs because we focus on small datasets on this table, for which MLPs are considered sufficient. There are actually some other arguments against DCGAN-based structure for synthetic tabular data generation, which is elaborated in Appendix A.1. CombU (Li et al., 2024) has been shown to be an effective activation function for tabular data generation. Therefore, we replace all activation layers, except for the output layer, with CombU.

## 3.3 TRAINING

### 3.3.1 TRAINING BY SAMPLING

For TAEGAN, we relax the constraint that only one and discrete component can be selected in $\mathbf{m}$ to any valid non-zero $\mathbf{m} \in \{0, 1\}^{D_m} \setminus \{\mathbf{0}\}$. Thus, $\boldsymbol{\mu}$ no longer remains as a one-hot vector, but a vector containing some data, but partially masked. We refer to this vector in TAEGAN as *hint vector* by its nature of being no longer always one-hot. Selected parts are considered known for the generator, which gives its name as it essentially gives *hint* to the MAE generator on what to recover and generate. The way masks are sampled is no longer simple random selection of columns by the varying number of components selected. Firstly, we need to determine the number of dimensions selected, which, instead of uniformly sampled from 1 to $n$, we sample by normalized inverse weights, namely, $1/i$ as the weight for $i$ components selected with the values normalized so that weights of all components sum to 1. Thus, smaller known dimensions are favored, in light of the fact that TAEGAN in essence a generative but not predictive network, so it should know less in general. Then, components are selected randomly based on the number of components selected derived above.

By making $\|\mathbf{m}\|_1 = 1$ no longer an invariant, the column-then-value sampling method of selecting $\boldsymbol{\gamma}$ becomes inapplicable. However, the same idea can be maintained if we describe the method in

---

**Algorithm 1:** TAEGAN Training Algorithm

---

**Data:** Training table $\mathcal{T}$, pre-training & main epochs $E_p, E$, network parameters $\boldsymbol{\theta}_G, \boldsymbol{\theta}_D$

**Result:** $\boldsymbol{\theta}_G$

Preprocess table $\mathcal{T}$ by one-hot encoding and VGM decomposition;
Calculate $\mathcal{W}$ to prepare for sampling;
**for** $e$ **in** $1, 2, \ldots, E_p + E$ **do**
   | **repeat**
   |   | $\mathbf{x}, \mathbf{m}, \boldsymbol{\mu}, \boldsymbol{\gamma} \leftarrow$ sampled data and masks ;              `/* Based on ` $\mathcal{W}$ `*/`
   |   | Generate $\mathbf{x}' \leftarrow G(\mathbf{m}, \boldsymbol{\gamma}, \mathbf{z})$ based on noise $\mathbf{z}$ from latent space;
   |   | Update $\boldsymbol{\theta}_G$ based on reconstruction loss of $\mathbf{x}'$ compared to $\mathbf{x}$;
   |   | Discriminate the data $y_r \leftarrow D(\mathbf{m}, \mathbf{x}), y_f \leftarrow D(\mathbf{m}, \mathbf{x}')$;
   |   | Update $\boldsymbol{\theta}_D$ based on classification loss for discrimination;
   |   | **if** $e > E_p$ **then**
   |   |   | `/* Do the adversarial generation after pre-training    */`
   |   |   | Update $\boldsymbol{\theta}_G$ based on the classification loss for making incorrect prediction;
   |   | **end**
   | **until** $M$ *times*;
**end**

---

another way. For each discrete component, a weight vector $\mathbf{w}_{dn}$ can be constructed by the following process, also seen in Figure 3b:

1. Collect the frequency of each value in the entire column. Let the frequency of value $v_{dn}$ be $f(v_{dn})$, $v_{dn}^{[i]}$ be the value in this component in $i$-th row (superscript is skipped if no ambiguity is present), and $\mathcal{V}_{dn}$ be the value range in this component.

2. Transform the frequencies to $g(v_{dn}) = \log(f(v_{dn}) + 1)$.

3. Normalize the frequencies as probability mass function values, such that $G_n = \sum_{v \in \mathcal{V}_{dn}} g(v)$, and $h(v_{dn}) = g(v_{dn})/G_n$. Assign $w_{dn}^{[i]} = h(v_{dn}^{[i]})/f(v_{dn}^{[i]})$.

Training-by-sampling is essentially randomly picking a discrete component and picking a row based on its corresponding weight vector. In TAEGAN, we propose a novel strategy – multivariate training-by-sampling. We relax the constraint that picked component must be discrete, by binning the continuous component so that they become discrete too. By putting the weight vectors of all components together, we can thus create a weight matrix $\mathcal{W} \in [0, 1]^{M \times D_m}$. Instead of randomly selecting one component to be the weight for row selection, TAEGAN calculates the mean over selected components in the mask to be the weight for row selection. Thus we extend training-by-sampling to multiple components. Nevertheless, we will ignore the interaction between components for the sampling process, as it is aimed at countering the column imbalance during model training. The overall multivariate training-by-sampling process is summarized in Figure 3c.

### 3.3.2 MODEL TRAINING PROCESS

There are three major tasks during the training of TAEGAN: generation, discrimination, as in any GAN, and reconstruction, which is defined for the auto-encoder. Discrimination task requires gradients only on the discriminator network, $D$, while the reconstruction task requires gradients on the generator, i.e., the auto-encoder network, $G$. Hence, the two networks may be pre-trained jointly before the adversarial generation starts. Then, the adversarial generation step can be added for the main training stage. The pre-training allows both the generator and discriminator networks to get some knowledge of the data, so that the adversarial training of GAN can be more stable. Pseudo-code for the process can be found in Algorithm 1.

A nuance we introduce is the noise construction. Typical GANs construct noise from standard Gaussian distribution. However, we contend that tabular data are usually naturally clustered, instead of perfectly distributed in Gaussian latent space. Therefore, instead of constructing a noise vector based on multivariate Gaussian distribution, we only construct half of the noise vector's dimensions using Gaussian distribution, and the rest are sampled randomly as either 0 or 1. Thus, half of the latent space becomes discrete, and hence may be used to imply clusters.

---

**Algorithm 2:** TAEGAN Sampling Algorithm

---

**Data:** Training table $\mathcal{T}$, generator parameters $\boldsymbol{\theta}_G$
**Result:** Generated row $\mathbf{x}'$
$\boldsymbol{\sigma} \leftarrow$ randomly generated order (permutation of $[1, 2, \ldots, D_m]$ with first $D_d$ being categorical indices);
$\mathbf{m} \leftarrow \mathbf{0}, \mathbf{m}_{\sigma_1} \leftarrow 1$ ;                       /* Initial mask with one selected only */
$\mathbf{x}' \leftarrow$ randomly sampled $\boldsymbol{\gamma}$ with the current $\mathbf{m}$;
**for** $i$ *in* $2, 3, \ldots, D_m$ **do**

    $\boldsymbol{\gamma} \leftarrow \boldsymbol{\mu}^T \cdot \mathbf{x}'$ with $\boldsymbol{\mu}$ constructed based on $\mathbf{m}$;

    Generate $\mathbf{x}' \leftarrow G(\mathbf{m}, \boldsymbol{\gamma}, \mathbf{z})$ based on noise $\mathbf{z}$ from latent space;

    $\mathbf{m}_{\sigma_i} \leftarrow 1$ ;                       /* Update mask with an additional component */
**end**

---

### 3.3.3 LOSS CONSTRUCTION

Losses for the discriminator and generator of CTAB-GAN+ (Zhao et al., 2024) will be maintained in TAEGAN, including WGAN-GP (Gulrajani et al., 2017) and an information loss based on the difference of statistics of the last layer of the discriminator between real and synthetically generated data (Park et al., 2018).

The cross entropy loss based on the condition (Xu et al., 2019) and classifier loss (Odena et al., 2017) will be replaced by the reconstruction loss, where cross entropy will be maintained for categorical components, and smooth L1 loss is used for continuous components. Consider the fact that the more masked data, the fewer components are known, the wider the range of the unknown components can be, and the less likely a predictor can be trained with good performance, it is hence unfair to make all reconstruction losses equally weighted for any mask. To solve the problem, we scale the reconstruction loss based on the proportion of components known, so that the reconstruction loss will be multiplied by $\|\mathbf{m}\|_1/|\mathbf{m}|$. To ensure a sufficiently large weight on the reconstruction loss when only a small number of components are selected, we introduce a minimum threshold $\tau$ for the weight. The weight will then be proportionally scaled from the range $[0, 1]$ to $[\tau, 1]$.

We also introduced a new interaction loss to the training process to let the model learn better inter-feature relation. By "interaction", we basically mean it in terms of statistics, a multiplication between different components. Each component is a 1D vector with varying sizes, so we calculate the outer products. We calculate the interaction between all components for each data row and flatten them, and we call the resulting vector the *interaction factor $\boldsymbol{\rho}$* of the row, which is calculated by $\boldsymbol{\rho} = \oplus_{i,j \in \{d1, d2, \ldots, dN, c1, c2, \ldots, cN\}, i \neq j}(\boldsymbol{\mu}_i \otimes \boldsymbol{\mu}_j)$, where $\otimes$ means outer product with flattening.

The loss is calculated by the mean root squared error of the means and standard deviations of all dimensions of the interaction factor between real and generated data on the batch. Note that it takes $O(N^2)$ to calculate the loss, so that it is not recommended for datasets with a large number of columns and efficiency is a concern. This way of calculating interaction is particularly useful for tabular data encoded with one-hot and VGM decomposition. For example, for two categorical components, the flattened outer product is another one-hot vector. Then, by forcing the mean and standard deviation of all dimensions in the outer products of synthetic data to be similar to the real ones, we can effectively avoid invalid category pairs, which is ubiquitous in real-life datasets, such as nationality and residency status within a certain country, or biological gender with compulsory national service completion. Since the modes for continuous columns are also represented in one-hot, this method is also able to find numeric correlations better if two columns have a high correlation.

### 3.4 SAMPLING

The data and mask sampling for generation is not based on logarithm-transformed frequency. It is essentially $\mathcal{W}$ with all values being $1/M$. Now that the mask is arbitrary as long as non-empty, a single row can be sampled in multiple steps, or in the extreme case, one component after one. Although each time all dimensions are predicted, we select only one additional and proceed to a next iteration. The full process is described in Algorithm 2.

This approach makes the generation of each component in a row sequential, which indeed slows down the overall sampling speed. Fortunately, due to the small model size, the inference speed of TAEGAN is still much faster than transformer-based methods such as REaLTabFormer (Solatorio & Dupriez, 2023). A detailed discussion is provided in Appendix 5 due to space constraints.

Note that the more components are known, the less variance is expected for a next component. Thus, we apply a varying temperature for prediction of discrete components, whose value divides the logits before passed into Gumbel-softmax activation. The temperature is designed to be linearly correlated to the ratio of known components, so higher temperature for less components known and vice versa.

# 4 EXPERIMENT RESULTS

## 4.1 EXPERIMENT SETUP

Data augmentation performance is evaluated through machine learning tasks, where the training set consists of both the original training data and synthetic data generated by the trained generator, with the synthetic data matching the size of the original training set. The test set is always composed of real data and remains untouched by the generator to prevent data leakage. The machine learning model used is XGBoost (Chen & Guestrin, 2016). Experiments are run on 8 commonly used benchmark tabular datasets on OpenML (Vanschoren et al., 2013) that have less than 2k rows, which we interpret as datasets with a need for augmentation. Dataset is summarized in Table 1. More details can be found in Appendix B.1.

We also evaluate the performance of synthetic data quality by classical machine learning efficacy based on train-on-synthetic-test-on-real framework. This metric is run on the 8 small datasets for data augmentation and 2 additional larger datasets. We compare the performance of TAEGAN with the state-of-the-art public[2] version for various types of models, ARF (Watson et al., 2023) for non-neural-network method, CTAB-GAN+ (Zhao et al., 2024) for GAN (Goodfellow et al., 2014), TVAE (Xu et al., 2019) for VAE (Kingma & Welling, 2013), TabDDPM (Kotelnikov et al., 2023) for Diffusion model (Ho et al., 2020), and REaLTabFormer (Solatorio & Dupriez, 2023) for large language models (LLMs). Implementation details of all models, including TAEGAN, are found in Appendix B.2.

| Dataset | Abbr. | Size | #C | #N | Aug? | Ab? |
|---|---|---|---|---|---|---|
| Breast cancer Wisconsin (original) (Street et al., 1993) | BW | 233 | 1 | 9 | X | |
| Blood transfusion service center (Yeh et al., 2009) | BT | 249 | 1 | 4 | X | |
| German credit (Hofmann, 1994) | CR | 333 | 14 | 7 | X | X |
| Diabetes (Kahn, 1994) | DI | 256 | 1 | 8 | X | X |
| Indian liver patient (Venkata Ramana et al., 2011) | IL | 194 | 2 | 9 | X | |
| QSAR biodegradation (Mansouri et al., 2013) | QS | 351 | 1 | 41 | X | |
| Spam e-mails (Cranor & LaMacchia, 1998) | SB | 1533 | 1 | 57 | X | |
| Breast cancer Winsconsin (diagnosis) (Street et al., 1993) | WD | 189 | 1 | 30 | X | |
| Adult (Kohavi, 1996) | AD | 16280 | 9 | 6 | | X |
| Covertype (Blackard, 1998) | CV | 16667 | 45 | 10 | | |

Table 1: Dataset summary. Size means the set of each split, i.e., train/val/test 1:1:1. "#C" and "#N" refer to the number of categorical and numeric columns, respectively. "Aug?" means whether the dataset is used for augmentation experiments, "Ab?" means whether the dataset is used for ablation experiments, and X means that the dataset is.

## 4.2 DATA AUGMENTATION COMPARED TO OTHER MODELS

Table 2 summarizes the augmentation effects for all synthetic data models. Not all models always guarantee the existence of data augmentation effect. However, TAEGAN indeed augmented data to improve the machine learning performance in all experimented datasets, and achieves the best data augmentation effect on 7 out of 8 of them compared to other synthetic data generation models.

In the next section, we will demonstrate that, in terms of other commonly used synthetic tabular data metrics, such as machine learning efficacy, TAEGAN does not achieve the same high success rate as it does for data augmentation, particularly when compared to LLM-based methods. The fact that TAEGAN (or GANs in general) performs particularly well for data augmentation on small datasets suggests that, while GANs may not capture internal data relationships as effectively as LLMs, they avoid being overskill, with model sizes only around 2-5% of LLM.

## 4.3 DATA QUALITY COMPARED TO OTHER MODELS

TAEGAN also achieves the best performance over a variety of datasets and metrics among all deep learning methods, but is not always the best outside this group. Table 3 gives the result of machine learning efficacy performances.

---

[2]"Public" by open-source code available. So, for example, although TabMT (Gulati & Roysdon, 2023) claims to be better than REaLTabFormer (Solatorio & Dupriez, 2023), it is not used.

| Dataset | Metric | Real | ARF | CTAB-GAN+ | TVAE | TabDDPM | REaLTabFormer | TAEGAN |
|---|---|---|---|---|---|---|---|---|
| BW | F1 | 0.974 | $0.965 \pm 0.004$ | $0.952 \pm 0.004$ | $\mathbf{0.977 \pm 0.004}$ | $0.958 \pm 0.009$ | $0.971 \pm 0.002$ | $\mathbf{0.977 \pm 0.002}$ |
|  | Acc | 0.974 | $0.975 \pm 0.002$ | $0.970 \pm 0.003$ | $0.975 \pm 0.001$ | $0.971 \pm 0.002$ | $0.975 \pm 0.001$ | $\mathbf{0.976 \pm 0.001}$ |
|  | Auc | 0.992 | $0.994 \pm 0.004$ | $0.990 \pm 0.004$ | $\mathbf{0.998 \pm 0.002}$ | $0.976 \pm 0.004$ | $\mathbf{0.998 \pm 0.002}$ | $\mathbf{0.998 \pm 0.002}$ |
| BT | F1 | 0.710 | $0.705 \pm 0.032$ | $0.712 \pm 0.011$ | $0.720 \pm 0.001$ | $\mathbf{0.725 \pm 0.007}$ | $0.705 \pm 0.014$ | $\mathbf{0.725 \pm 0.009}$ |
|  | Acc | 0.720 | $0.724 \pm 0.030$ | $0.733 \pm 0.012$ | $\mathbf{0.760 \pm 0.006}$ | $0.739 \pm 0.009$ | $0.725 \pm 0.013$ | $0.737 \pm 0.008$ |
|  | Auc | 0.630 | $0.634 \pm 0.018$ | $0.641 \pm 0.011$ | $0.619 \pm 0.003$ | $0.631 \pm 0.007$ | $0.623 \pm 0.014$ | $\mathbf{0.665 \pm 0.012}$ |
| CR | F1 | 0.669 | $0.687 \pm 0.013$ | $0.670 \pm 0.006$ | $0.681 \pm 0.008$ | $0.708 \pm 0.009$ | $0.696 \pm 0.010$ | $\mathbf{0.710 \pm 0.003}$ |
|  | Acc | 0.686 | $0.809 \pm 0.012$ | $0.681 \pm 0.004$ | $0.710 \pm 0.007$ | $0.724 \pm 0.009$ | $0.713 \pm 0.010$ | $\mathbf{0.728 \pm 0.003}$ |
|  | Auc | 0.701 | $0.690 \pm 0.036$ | $0.695 \pm 0.003$ | $0.707 \pm 0.005$ | $0.727 \pm 0.015$ | $0.703 \pm 0.011$ | $\mathbf{0.735 \pm 0.009}$ |
| DI | F1 | 0.747 | $0.728 \pm 0.005$ | $0.733 \pm 0.021$ | $\mathbf{0.761 \pm 0.006}$ | $0.746 \pm 0.019$ | $0.731 \pm 0.008$ | $0.759 \pm 0.004$ |
|  | Acc | 0.746 | $0.727 \pm 0.005$ | $0.746 \pm 0.019$ | $0.756 \pm 0.007$ | $0.749 \pm 0.018$ | $0.728 \pm 0.008$ | $\mathbf{0.758 \pm 0.003}$ |
|  | Auc | 0.798 | $0.797 \pm 0.010$ | $0.796 \pm 0.010$ | $\mathbf{0.823 \pm 0.004}$ | $0.804 \pm 0.008$ | $0.792 \pm 0.018$ | $\mathbf{0.823 \pm 0.006}$ |
| IL | F1 | 0.673 | $0.672 \pm 0.006$ | $0.631 \pm 0.024$ | $0.659 \pm 0.019$ | $0.676 \pm 0.016$ | $0.655 \pm 0.022$ | $\mathbf{0.702 \pm 0.005}$ |
|  | Acc | 0.682 | $0.665 \pm 0.006$ | $0.670 \pm 0.019$ | $0.685 \pm 0.017$ | $0.689 \pm 0.016$ | $0.670 \pm 0.019$ | $\mathbf{0.701 \pm 0.005}$ |
|  | Auc | 0.697 | $0.696 \pm 0.008$ | $0.658 \pm 0.036$ | $0.682 \pm 0.014$ | $0.697 \pm 0.004$ | $0.682 \pm 0.015$ | $\mathbf{0.706 \pm 0.011}$ |
| QS | F1 | 0.819 | $0.833 \pm 0.011$ | $0.839 \pm 0.003$ | $0.840 \pm 0.007$ | $0.840 \pm 0.004$ | $0.833 \pm 0.005$ | $\mathbf{0.845 \pm 0.005}$ |
|  | Acc | 0.818 | $0.831 \pm 0.011$ | $0.839 \pm 0.003$ | $0.839 \pm 0.006$ | $0.838 \pm 0.005$ | $0.845 \pm 0.005$ | $\mathbf{0.846 \pm 0.005}$ |
|  | Auc | 0.892 | $0.893 \pm 0.002$ | $\mathbf{0.909 \pm 0.002}$ | $0.892 \pm 0.006$ | $0.893 \pm 0.007$ | $0.903 \pm 0.002$ | $0.905 \pm 0.007$ |
| SB | F1 | 0.939 | $\mathbf{0.941 \pm 0.004}$ | $0.935 \pm 0.003$ | $0.934 \pm 0.001$ | $0.939 \pm 0.001$ | $0.937 \pm 0.002$ | $\mathbf{0.941 \pm 0.004}$ |
|  | Acc | 0.939 | $\mathbf{0.941 \pm 0.004}$ | $0.935 \pm 0.003$ | $0.934 \pm 0.001$ | $0.939 \pm 0.001$ | $0.936 \pm 0.001$ | $\mathbf{0.941 \pm 0.004}$ |
|  | Auc | 0.980 | $\mathbf{0.982 \pm 0.001}$ | $0.980 \pm 0.001$ | $\mathbf{0.982 \pm 0.001}$ | $0.980 \pm 0.001$ | $\mathbf{0.982 \pm 0.001}$ | $0.981 \pm 0.001$ |
| WD | F1 | 0.953 | $0.954 \pm 0.00$ | $0.953 \pm 0.013$ | $0.953 \pm 0.007$ | $\mathbf{0.964 \pm 0.013}$ | $0.962 \pm 0.011$ | $0.957 \pm 0.006$ |
|  | Acc | 0.953 | $0.955 \pm 0.013$ | $0.953 \pm 0.007$ | $0.949 \pm 0.007$ | $\mathbf{0.963 \pm 0.013}$ | $0.951 \pm 0.011$ | $0.957 \pm 0.007$ |
|  | Auc | 0.991 | $0.989 \pm 0.002$ | $0.989 \pm 0.004$ | $0.992 \pm 0.001$ | $0.992 \pm 0.001$ | $0.987 \pm 0.003$ | $\mathbf{0.994 \pm 0.001}$ |

Table 2: Augmentation performances of different synthetic tabular data generation models. All tasks are binary classification tasks, and weighted F1, accuracy (Acc), and ROC AUC (AUC) scores are calculated. Best performances are in bold.

The results show TAEGAN's superiority among deep learning methods. However, among all models, LLM-based REaLTabFormer (Solatorio & Dupriez, 2023) tends to be better, especially on larger datasets. This is unsurprising as LLMs naturally learn better with abundance of data, and LLMs has a few tens of the number of parameters than deep learning methods. REaLTabFormer for these datasets typically constructs a neural network with about 40M parameters, while TAEGAN's network has only 1-2M parameters. The results of some other metrics can be found in Appendix B.4.

### 4.4 ABLATION STUDY

For ablation study, we focus on data quality by machine learning efficacy, and base our experiments on three more commonly used benchmark datasets: AD, CR, and DI. The results are shown in Table 4. The experiments verify the effect of all optimizations introduced in TAEGAN.

## 5 CONCLUSION AND FUTURE WORK

### 5.1 CONCLUSION

In this paper, we proposed TAEGAN, which uses MAE as the generator of GAN to generate synthetic tabular data that leverages a solid self-supervised pre-training task for the data generation task to improve the performance. This novel GAN improves existing tabular GANs significantly. Moreover, TAEGAN shows a clear advantage in data augmentation compared to other synthetic tabular data generation models, and generates data with the highest quality among all non-LLM deep-learning methods.

### 5.2 FUTURE WORK

This paper discusses the ability in data augmentation of synthetic data generators. It can be combined with other existing synthetic data augmentation techniques to augment data further.

The flexible masking in the generator of TAEGAN makes the generator alone possible to be applied directly on downstream classification, regression, or imputation tasks. Also, the generator and dis-

| Dataset | Metric | Real | non-deep-learning | | deep-learning | | | |
|---|---|---|---|---|---|---|---|---|
| | | | ARF | REaLTabFormer | CTAB-GAN+ | TVAE | TabDDPM | TAEGAN |
| BW | F1 | 0.974 | $0.937 \pm 0.020$ | $\mathbf{0.979 \pm 0.004}$ | $0.923 \pm 0.020$ | $\mathbf{0.977 \pm 0.004}$ | $0.703 \pm 0.050$ | $\mathbf{0.977 \pm 0.002}$ |
| | Acc | 0.974 | $0.960 \pm 0.020$ | $\mathbf{0.979 \pm 0.004}$ | $0.928 \pm 0.019$ | $0.958 \pm 0.015$ | $0.761 \pm 0.030$ | $\mathbf{0.976 \pm 0.002}$ |
| | Auc | 0.992 | $0.988 \pm 0.010$ | $\mathbf{0.993 \pm 0.001}$ | $0.970 \pm 0.004$ | $0.990 \pm 0.004$ | $0.967 \pm 0.015$ | $\mathbf{0.993 \pm 0.000}$ |
| BT | F1 | 0.710 | $0.692 \pm 0.020$ | $0.703 \pm 0.026$ | $0.640 \pm 0.016$ | $0.702 \pm 0.000$ | $0.650 \pm 0.008$ | $\mathbf{0.706 \pm 0.018}$ |
| | Acc | 0.720 | $0.705 \pm 0.020$ | $0.724 \pm 0.020$ | $\mathbf{0.744 \pm 0.006}$ | $0.680 \pm 0.000$ | $0.697 \pm 0.025$ | $0.715 \pm 0.026$ |
| | Auc | 0.630 | $0.644 \pm 0.034$ | $0.610 \pm 0.038$ | $0.587 \pm 0.051$ | $0.621 \pm 0.000$ | $0.527 \pm 0.078$ | $\mathbf{0.676 \pm 0.013}$ |
| CR | F1 | 0.669 | $0.632 \pm 0.037$ | $\mathbf{0.683 \pm 0.001}$ | $0.211 \pm 0.037$ | $0.568 \pm 0.003$ | $0.574 \pm 0.030$ | $\mathbf{0.682 \pm 0.014}$ |
| | Acc | 0.686 | $0.659 \pm 0.038$ | $\mathbf{0.700 \pm 0.010}$ | $0.335 \pm 0.017$ | $0.672 \pm 0.000$ | $0.612 \pm 0.073$ | $\mathbf{0.691 \pm 0.021}$ |
| | Auc | 0.701 | $0.607 \pm 0.058$ | $0.675 \pm 0.014$ | $0.529 \pm 0.053$ | $0.588 \pm 0.012$ | $0.527 \pm 0.012$ | $\mathbf{0.723 \pm 0.017}$ |
| DI | F1 | 0.747 | $0.695 \pm 0.008$ | $0.727 \pm 0.011$ | $0.622 \pm 0.014$ | $0.725 \pm 0.023$ | $0.507 \pm 0.081$ | $\mathbf{0.791 \pm 0.018}$ |
| | Acc | 0.746 | $0.703 \pm 0.017$ | $0.724 \pm 0.010$ | $0.695 \pm 0.003$ | $0.721 \pm 0.021$ | $0.680 \pm 0.054$ | $\mathbf{0.789 \pm 0.018}$ |
| | Auc | 0.798 | $0.747 \pm 0.009$ | $0.785 \pm 0.018$ | $0.717 \pm 0.036$ | $0.780 \pm 0.020$ | $0.668 \pm 0.142$ | $\mathbf{0.859 \pm 0.010}$ |
| IL | F1 | 0.673 | $0.644 \pm 0.037$ | $0.635 \pm 0.014$ | $0.625 \pm 0.022$ | $0.619 \pm 0.034$ | $0.580 \pm 0.000$ | $\mathbf{0.647 \pm 0.025}$ |
| | Acc | 0.682 | $0.644 \pm 0.041$ | $0.662 \pm 0.015$ | $0.699 \pm 0.016$ | $\mathbf{0.707 \pm 0.015}$ | $0.703 \pm 0.000$ | $0.668 \pm 0.017$ |
| | Auc | 0.697 | $0.666 \pm 0.031$ | $0.620 \pm 0.004$ | $0.630 \pm 0.040$ | $\mathbf{0.687 \pm 0.022}$ | $0.581 \pm 0.042$ | $0.680 \pm 0.025$ |
| QS | F1 | 0.819 | $0.802 \pm 0.011$ | $\mathbf{0.839 \pm 0.010}$ | $0.622 \pm 0.032$ | $0.722 \pm 0.001$ | $0.622 \pm 0.018$ | $\mathbf{0.810 \pm 0.002}$ |
| | Acc | 0.818 | $0.805 \pm 0.011$ | $\mathbf{0.828 \pm 0.009}$ | $0.702 \pm 0.014$ | $0.741 \pm 0.006$ | $0.655 \pm 0.015$ | $\mathbf{0.812 \pm 0.002}$ |
| | Auc | 0.892 | $0.865 \pm 0.011$ | $\mathbf{0.894 \pm 0.006}$ | $0.736 \pm 0.009$ | $0.745 \pm 0.005$ | $0.712 \pm 0.008$ | $\mathbf{0.879 \pm 0.005}$ |
| SB | F1 | 0.939 | $0.903 \pm 0.005$ | $\mathbf{0.905 \pm 0.003}$ | $0.754 \pm 0.019$ | $0.679 \pm 0.011$ | $0.598 \pm 0.001$ | $\mathbf{0.899 \pm 0.002}$ |
| | Acc | 0.939 | $0.904 \pm 0.005$ | $\mathbf{0.904 \pm 0.003}$ | $0.779 \pm 0.014$ | $0.678 \pm 0.010$ | $0.618 \pm 0.002$ | $\mathbf{0.900 \pm 0.002}$ |
| | Auc | 0.980 | $0.963 \pm 0.002$ | $\mathbf{0.967 \pm 0.001}$ | $0.932 \pm 0.005$ | $0.835 \pm 0.001$ | $0.745 \pm 0.010$ | $\mathbf{0.955 \pm 0.001}$ |
| WD | F1 | 0.953 | $0.917 \pm 0.014$ | $0.907 \pm 0.009$ | $0.926 \pm 0.013$ | $0.909 \pm 0.005$ | $0.913 \pm 0.008$ | $\mathbf{0.933 \pm 0.002}$ |
| | Acc | 0.953 | $0.918 \pm 0.014$ | $0.919 \pm 0.018$ | $0.927 \pm 0.013$ | $0.907 \pm 0.005$ | $0.922 \pm 0.015$ | $\mathbf{0.934 \pm 0.003}$ |
| | Auc | 0.991 | $0.979 \pm 0.006$ | $0.927 \pm 0.016$ | $0.971 \pm 0.008$ | $0.983 \pm 0.005$ | $0.926 \pm 0.015$ | $\mathbf{0.984 \pm 0.002}$ |
| AD | F1 | 0.864 | $0.844 \pm 0.002$ | $\mathbf{0.857 \pm 0.002}$ | $0.820 \pm 0.001$ | $0.816 \pm 0.002$ | $0.822 \pm 0.005$ | $\mathbf{0.844 \pm 0.001}$ |
| | Acc | 0.868 | $0.850 \pm 0.002$ | $\mathbf{0.861 \pm 0.002}$ | $0.833 \pm 0.000$ | $0.815 \pm 0.001$ | $0.838 \pm 0.003$ | $\mathbf{0.850 \pm 0.001}$ |
| | Auc | 0.921 | $0.900 \pm 0.002$ | $\mathbf{0.912 \pm 0.001}$ | $0.885 \pm 0.001$ | $0.868 \pm 0.003$ | $0.888 \pm 0.002$ | $\mathbf{0.900 \pm 0.001}$ |
| CV | F1 | 0.919 | $0.836 \pm 0.001$ | $\mathbf{0.898 \pm 0.001}$ | $0.746 \pm 0.005$ | $0.782 \pm 0.006$ | $0.768 \pm 0.002$ | $\mathbf{0.811 \pm 0.000}$ |
| | Acc | 0.919 | $0.840 \pm 0.001$ | $\mathbf{0.898 \pm 0.001}$ | $0.756 \pm 0.006$ | $0.790 \pm 0.006$ | $0.784 \pm 0.005$ | $\mathbf{0.817 \pm 0.000}$ |
| | Auc | 0.993 | $0.977 \pm 0.001$ | $\mathbf{0.989 \pm 0.000}$ | $0.937 \pm 0.001$ | $0.954 \pm 0.001$ | $0.940 \pm 0.007$ | $\mathbf{0.969 \pm 0.000}$ |

Table 3: Data quality by machine learning performance of train-on-synthetic-test-on-real strategy. All tasks are binary classification tasks except for CV, which is multi-class classification, and weighted F1, accuracy (Acc), and ROC AUC (AUC) scores (OVR for multi-class) are calculated. Best performances among all models and among deep learning methods are both in bold.

| Description | AD | CR | DI |
|---|---|---|---|
| All training steps being main (pre- and main training steps by ratio 10:3) | $0.843 \pm 0.001$ | $0.645 \pm 0.011$ | $0.746 \pm 0.025$ |
| No interaction loss (has interaction loss) | $0.843 \pm 0.002$ | $0.547 \pm 0.034$ | $0.732 \pm 0.024$ |
| All continuous noise (half being discrete) | $0.844 \pm 0.003$ | $0.670 \pm 0.023$ | $0.779 \pm 0.007$ |
| Uniform sampling during training (multivariate training-by-sampling with logarithmic transformation) | $0.839 \pm 0.002$ | $0.644 \pm 0.011$ | $0.733 \pm 0.004$ |
| Constant learning rate (exponentially decaying learning rate) | $0.840 \pm 0.001$ | $0.679 \pm 0.011$ | $0.776 \pm 0.017$ |
| TAEGAN | $0.844 \pm 0.001$ | $0.682 \pm 0.014$ | $0.791 \pm 0.018$ |

Table 4: ML efficacy performances across different datasets for ablation study. Each ablation experiment's description has the described change only, and the content in bracket means the actual setting in TAEGAN for easier reference. The reported scores are weighted F1 for XGBoost classification.

criminator themselves can be used as auxiliary pre-training networks to improve the performance in downstream tasks.

Many ideas in this paper can also be extended to other use cases. For example, the multivariate training-by-sampling method can be useful for addressing skewness in any tabular-like data task.

Additionally, TAEGAN's framework could potentially be applied beyond tabular data to other modalities, such as images, by incorporating appropriate backbones for the generator (including its encoder and decoder) and discriminator.

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

## A    ANALYSIS OF THE PROS AND CONS OF SOME MODELS

### A.1    CTAB-GAN+

GANs are relatively light neural networks (light compared to LLMs), and hence although they do not generate better-quality data than LLMs, they have much utility value in reality, as they generate decent-quality data with only a small proportion of the time needed for LLMs.

CTAB-GAN+ (Zhao et al., 2024) inherits the neural network architecture design of TableGAN (Park et al., 2018), which uses DCGAN (Radford et al., 2016) as the backbone. While CNN-based DC-GAN works well for images, using pure CNNs for both the generator and discriminator is actually problematic. Much of CNNs' weights are localized regardless of the position. This is catered for the property of images, whose position matters much less than the content in whatever position. However, the "images" in tabular GANs using DCGAN backbone is simply a matrix constructed by reshaping the encoded vector of each data row. Thus, each position in the "image" has specific meaning in the actual tabular data because it corresponds to some specific encoded information of a specific column. Therefore, CNNs that do not emphasize on positions is actually not suitable for the case. Even if ultimately tabular data may have different columns sharing same internal pattern, the removal of position should not happen immediately at input. This is also one of the reasons TAEGAN switched the backbone back to simple MLPs.

If one still wants to use CNNs for tabular GANs, one suggestion would be at least appending some fully connected layers at the end of the generator and start of discriminator. This would also solve the memory issue of the current design of CTAB-GAN+ for datasets with a large number of columns. The current design of CTAB-GAN+ has the image sizes, and hence neural network sizes, highly determined by the size of the encoded vector for each data row, and datasets with a few hundreds of rows immediately makes the networks huge, and even batch one makes it out-of-memory on normal single GPU machines.

### A.2    ARF

ARF (Watson et al., 2023) is a CPU-friendly non-neural-network synthetic tabular data generator, which generates high quality data very efficiently. However, it suffers from privacy concerns, especially when the model is used for privacy-preserving data sharing. In ARF, the exact marginal distribution in all columns are calculated on each terminal node in each tree, so when the tree leaf size is small enough, in the extreme case, 1, then the real data can be perfectly reconstructed. Fortunately, ARF limits the minimum leaf size, to 5 by default. However, a vast majority of the leaf nodes will have sizes very close to this minimum limit, and nodes along the tree branches also determines the values of relevant features for the corresponding tree leaves. Consequently, the real data can be recovered to a large extent, including some real records to be recovered completely. By analyzing some datasets and fitted ARF structure, about 30% of the features on average can be completely determined without ambiguity, including numeric values up to the exact precision, by the sampled terminal's branch and tree index, and about 3-10% of real data records can be generated exactly.

### A.3    LLM-BASED MODELS

LLMs generates the best-quality data among all models, which is unsurprising given the success of transformer in almost all different kind of tasks. The sampling algorithm of tokens also effectively reduces the probability of exact matches of real data being produced.

However, due to the large size of LLMs, this group of models typically take much longer than other models to train. Some papers of these models even explicitly mentions the inefficiency problem (Gulati & Roysdon, 2023) of the model.

Also, it has always been agreed that LLMs suffer from a potential of memorizing the training data (Hartmann et al., 2023). This problem also inherits to LLMs when it is used in the realm of tabular data, as Borisov et al. (2023) mentions that naïve application of LLMs on textualized tabular data, when trained for large epochs, has a high potential of memorizing the training data and producing them, and Bordt et al. (2024) also establishes a comprehensive analysis on the risk of using LLMs for tabular data synthesis in terms of privacy.

REaLTabFormer (Solatorio & Dupriez, 2023) is an LLM model for tabular data generation that designs mechanisms to purposely mitigate data memorizing and overfitting problem. However, it still has problems regarding privacy. On the datasets we experimented, we performed membership inference attacks, which trains a classifier with the training data for the generator as class 1 and a validation dataset of the same size as class 0, and tries to classify the synthetic data. The result is that a significant majority of the synthetic data is classified as class 1, which means that the model is not safe under membership inference attack. Also, by manipulating the sampling parameter, such as reducing temperature and beam search with diversity requirement, the trained model may still be exploited to recover the real data.

In addition, LLMs suffer from similar out-of-memory issues as CTAB-GAN+, as the sequence lengths for each sentence is completely determined by the number of columns and their representation lengths. Although backbones of LLM-based models can always be swapped to use the most up-to-date and suitable ones, so that models with larger context can be used in place, it might not be a good choice. After all, these models are trained on a tabular dataset, and having a real LLM-sized model could be significant over-skill.

In Section 3.4, we also mentioned that language model's sampling process is slow, so that repeated sampling by gradually removing the masks takes a long time. We show the sampling time on the two larger datasets in Table 5. TAEGAN samples values for each column one at a time and gradually removes the mask, and REaLTabFormer samples all values at once. The sampling time of it multiplied by the total number of columns are also shown in the table, and its inefficiency is clearly demonstrated compared against TAEGAN. This inefficiency problem would be particularly a problem with REaLTabFormer with iterative masked generation when the number of columns are large. The number of columns not only result in an O(#C) additional sampling time, but also explodes the sequence lengths, which usually requires the model size to increase in $O(\#C^2)$ due to the existence of attention layers in transformers (Vaswani et al., 2017).

| Dataset | REaLTabFormer | REaLTabFormer $\times$ #C | TAEGAN |
|---|---|---|---|
| AD | $21.401 \pm 0.028$ | 321.015 | $4.428 \pm 0.004$ |
| CV | $66.134 \pm 0.564$ | 3637.370 | $32.802 \pm 0.345$ |

Table 5: Sampling time (unit in second) of TAEGAN versus REaLTabFormer on larger datasets. REaLTabFormer $\times$ #C means the the time for REaLTabFormer multiplied by the total number of columns in the dataset, showing the estimated time when it is also generated iteratively with masks removed gradually.

# B DETAILS OF EXPERIMENTS

## B.1 EXPERIMENT DATASETS

All datasets are obtained from OpenML (Vanschoren et al., 2013) by `sklearn.datasets.from_openml(HANDLE)`. All datasets are randomly split into $1 : 1 : 1$ for training, validation, and testing sets. Table 6 summarizes the information of all datasets used. All datasets are binary classification sets except for CV, which is multi-class with 7 classes.

AD dataset originally contains some N/A values. We fill numeric N/A values with -9999 and categorical N/A values with a new N/A category. CV dataset originally is much larger, but because this paper focuses more on smaller dataset, so we sample some records to do experiments.

| Dataset (`HANDLE`) | Abbr. | Size | #C | #N | Aug? | Ab? |
|---|---|---|---|---|---|---|
| Breast cancer Wisconsin (original) (Street et al., 1993) (`breast-w`) | BW | 233 | 1 | 9 | X | |
| Blood transfusion service center (Yeh et al., 2009) (`blood-transfusion-service-center`) | BT | 249 | 1 | 4 | X | |
| German credit (Hofmann, 1994) (`credit-g`) | CR | 333 | 14 | 7 | X | X |
| Diabetes (Kahn, 1994) (`diabetes`) | DI | 256 | 1 | 8 | X | X |
| Indian liver patient (Venkata Ramana et al., 2011) (`ilpd`) | IL | 194 | 2 | 9 | X | |
| QSAR biodegradation (Mansouri et al., 2013) (`qsar-biodeg`) | QS | 351 | 1 | 41 | X | |
| Spam e-mails (Cranor & LaMacchia, 1998) (`spambase`) | SB | 1533 | 1 | 57 | X | |
| Breast cancer Winsconsin (diagnosis) (Street et al., 1993) (`wdbc`) | WD | 189 | 1 | 30 | X | |
| Adult (Kohavi, 1996) (`adult`) | AD | 16280 | 9 | 6 | | X |
| Covertype (Blackard, 1998) (`covertype`) | CV | 16667 | 45 | 10 | | |

Table 6: Details of datasets used for evaluation. Size means the set of each split. "#C" and "#N" refers to the number of categorical and numeric columns, inclusive of target, respectively. "Aug?" means whether the dataset is used for augmentation experiments, "Ab?" means whether the dataset is used for ablation experiments, and an X means that the dataset is.

## B.2    DETAILED EXPERIMENT SETUP

### B.2.1    GENERAL EXPERIMENT SETUP

For each dataset's experiment, the generators are traiend on the training set only. Each generative model, after training, will be used to generate 3 synthetic datasets with the same size as the actual training set without validation set, and the metrics are collected from the three copies of synthetic datasets.

Experiments are done on an NVIDIA GeForce RTX 4090 GPU.

### B.2.2    TAEGAN TRAINING SETUP

The encoder and decoder of the auto-encoder generator, as well as the discriminator, are all 6-level MLPs with 0.2 dropout rate, layer norms, and CombU (Li et al., 2024) activation. The size of hidden layers and embedding layer between encoder and decoder are all 256. The noise dimension is 128.

The model is trained with 90 warmup epochs and 300 epochs with batch size 500. The learning rate is exponentially decaying starting from $2 \times 10^{-3}$ with decaying rate being 0.99 per epoch. The weight decay for the Adam optimizer of the networks is $1 \times 10^{-5}$. In actual implementation of each step, discrimination step is executed twice, adversarial generation step is executed once, and reconstruction step is calculated twice.

### B.2.3    OTHER MODELS' SETUP

We use the default settings from each model as per published in their most recent code, with the correct categorical versus numeric data types based on the definition of the datasets.

We also do evaluation on the synthetic data using a few other metrics, and we will report the results in this section.

### B.3 EVALUATION METRICS

### B.3.1 SUPPLEMENTARY METRICS

Some supplementary metrics are also used to evaluate the quality of the synthetic data:

- **Marginal Distribution:** Jensen-Shannon divergence (Lin, 1991) is calculated, where continuous columns are binned by quantiles into 20 bins.

- **Correlation Difference:** The difference in the correlation matrix, where Pearson's R is used for continuous-continuous cases (Pearson, 1895), correlation ratio is used for categorical-continuous cases (Pearson, 1905), and Cramér's V for categorical-categorical cases (Cramér, 1999).

- **Discrimination:** Same thing as the discriminator in GAN, but it is not a neural network and is used for evaluation. The better the performance of a discriminator can be, the worse the synthetic data's quality is. We use XGBoost (Chen & Guestrin, 2016) classifier. The training data comes from the training data of the generative model as real data, and a training set from $4:1$ split of the generated data as fake data. To make the score fairer across different datasets, we randomly select samples among each of them so that their size is capped by the smaller size of the two. The test set for the discrimination metric comes from the real test data set and another split from the generated data, also randomly sampled by a size capped by the smaller size of the two. Accuracy, F1, and ROC AUC scores are reported.

### B.3.2 METRIC AGGREGATION

Since there are many different values for the metrics above can be calculated for the same dataset by the existence of multiple columns, we apply some aggregation to show the result. When aggregating the scores, duplicated scores (for example, the correlation of column 1 to column 2 and the correlation of column 2 to column 1) are not counted twice.

A commonly agreed understanding of scores is a value in $[0, 1]$ with a maximum objective. We also normalize scores into this scale before aggregation. Jensen-Shannon divergence has fixed range $[0, 1]$, and the correlation difference (absolute value) has range $[0, 1]$. By simple scaling, linear translation, and additive inverse, they can both be easily transformed to scores in range $[0, 1]$ with a maximum optimal objective. Then, calculate the harmonic mean over all scores to be the score for the metric. Harmonic instead of arithmetic mean is used for the higher sensitivity of harmonic means to a single bad score, which typically features a mode of failure in the synthetic data. Since some models are stronger at categorical columns and some others are stronger at numeric columns, we do not aggregate the scores of categorical and numeric together. Discrimination score need not be aggregated because only one score is calculated. Note that as long as one value is 0, the harmonic mean would become zero, which may make the scores not informative if everything becomes 0. Thus, we scale the values in $[0.1, 1]$ before calculating the harmonic mean.

For marginal distribution and correlation difference, real dataset compared to itself would always give a perfect 0 value. However, another sampled dataset may not maintain a perfect 0 value due to the randomness. Thus, the scores need to be normalized by the score of a separate validation set that is not seen by the generative model to extend the meaning from fidelity to the training dataset to its generalizability. After the scores are calculated, we normalize the scores by the score of the validation set by $S' = 1 - \hat{S} \cdot |S - \hat{S}|$, where $S, \hat{S}, S'$ are the raw score, validation score, and normalized scores respectively. This way, the score is still controlled strictly in the range $[0, 1]$ with a maximum optional objective, but higher normalized score if the validation score is not good, given the same raw score.

### B.4 SUPPLEMENTARY METRIC RESULTS

Table 7-8 gives the marginal distribution scores and correlation scores for the 10 datasets. Almost all models are able to capture marginal distribution and correlation relatively good for all datasets. ARF and REaLTabFormer are particularly good in marginal distribution, which is unsurprising as ARF keeps the exact distribution in all tree nodes, and REaLTabFormer uses the sampling method

| Dataset | Metric | non-deep-learning | | deep-learning | | | |
|---------|--------|------|--------------|-----------|------|---------|--------|
| | | ARF | REaLTabFormer | CTAB-GAN+ | TVAE | TabDDPM | TAEGAN |
| BW | C | $0.997 \pm 0.001$ | $0.999 \pm 0.001$ | $0.990 \pm 0.001$ | $0.969 \pm 0.002$ | $0.926 \pm 0.005$ | $0.973 \pm 0.002$ |
| | D | $1.000 \pm 0.000$ | $0.998 \pm 0.002$ | $1.000 \pm 0.000$ | $0.988 \pm 0.005$ | $1.000 \pm 0.000$ | $1.000 \pm 0.000$ |
| BT | C | $0.997 \pm 0.002$ | $0.991 \pm 0.001$ | $0.995 \pm 0.003$ | $0.973 \pm 0.003$ | $0.931 \pm 0.006$ | $0.988 \pm 0.002$ |
| | D | $1.000 \pm 0.000$ | $1.000 \pm 0.000$ | $0.973 \pm 0.008$ | $0.925 \pm 0.000$ | $1.000 \pm 0.000$ | $0.999 \pm 0.001$ |
| CR | C | $0.998 \pm 0.001$ | $0.996 \pm 0.000$ | $0.992 \pm 0.002$ | $0.939 \pm 0.002$ | $0.927 \pm 0.003$ | $0.993 \pm 0.001$ |
| | D | $0.999 \pm 0.001$ | $1.000 \pm 0.000$ | $0.911 \pm 0.003$ | $0.909 \pm 0.001$ | $0.945 \pm 0.003$ | $1.000 \pm 0.000$ |
| DI | C | $0.997 \pm 0.001$ | $0.993 \pm 0.001$ | $0.998 \pm 0.001$ | $0.970 \pm 0.002$ | $0.902 \pm 0.002$ | $0.991 \pm 0.002$ |
| | D | $1.000 \pm 0.000$ | $1.000 \pm 0.000$ | $0.980 \pm 0.003$ | $0.996 \pm 0.001$ | $0.999 \pm 0.000$ | $0.999 \pm 0.001$ |
| IL | C | $0.995 \pm 0.004$ | $0.992 \pm 0.001$ | $0.982 \pm 0.003$ | $0.946 \pm 0.005$ | $0.959 \pm 0.001$ | $0.967 \pm 0.006$ |
| | D | $1.000 \pm 0.000$ | $1.000 \pm 0.000$ | $0.923 \pm 0.009$ | $0.955 \pm 0.007$ | $0.953 \pm 0.002$ | $0.998 \pm 0.001$ |
| QS | C | $0.993 \pm 0.000$ | $0.999 \pm 0.000$ | $0.985 \pm 0.001$ | $0.934 \pm 0.001$ | $0.978 \pm 0.001$ | $0.982 \pm 0.001$ |
| | D | $0.998 \pm 0.001$ | $0.998 \pm 0.000$ | $0.971 \pm 0.003$ | $0.973 \pm 0.000$ | $0.997 \pm 0.000$ | $0.998 \pm 0.001$ |
| SB | C | $0.993 \pm 0.000$ | $0.995 \pm 0.001$ | $0.996 \pm 0.000$ | $0.967 \pm 0.000$ | $0.932 \pm 0.001$ | $0.995 \pm 0.000$ |
| | D | $1.000 \pm 0.000$ | $1.000 \pm 0.000$ | $0.984 \pm 0.001$ | $1.000 \pm 0.000$ | $1.000 \pm 0.000$ | $1.000 \pm 0.000$ |
| WD | C | $0.997 \pm 0.002$ | $0.946 \pm 0.001$ | $0.989 \pm 0.001$ | $0.977 \pm 0.003$ | $0.976 \pm 0.001$ | $0.977 \pm 0.001$ |
| | D | $0.999 \pm 0.001$ | $0.983 \pm 0.006$ | $0.999 \pm 0.000$ | $0.998 \pm 0.001$ | $1.000 \pm 0.000$ | $0.999 \pm 0.002$ |
| AD | C | $0.993 \pm 0.000$ | $0.998 \pm 0.000$ | $0.998 \pm 0.000$ | $0.972 \pm 0.000$ | $0.930 \pm 0.000$ | $0.991 \pm 0.000$ |
| | D | $1.000 \pm 0.000$ | $0.998 \pm 0.000$ | $0.954 \pm 0.000$ | $0.989 \pm 0.000$ | $0.917 \pm 0.000$ | $0.997 \pm 0.000$ |
| CV | C | $0.997 \pm 0.000$ | $1.000 \pm 0.000$ | $0.996 \pm 0.000$ | $0.978 \pm 0.000$ | $0.967 \pm 0.000$ | $0.981 \pm 0.000$ |
| | D | $1.000 \pm 0.000$ | $1.000 \pm 0.000$ | $0.996 \pm 0.000$ | $0.999 \pm 0.000$ | $0.991 \pm 0.000$ | $0.999 \pm 0.000$ |

Table 7: Normalized marginal distribution of each model. "D" stands for discrete, and "C" stands for continuous.

| Dataset | Metric | non-deep-learning | | deep-learning | | | |
|---------|--------|------|--------------|-----------|------|---------|--------|
| | | ARF | REaLTabFormer | CTAB-GAN+ | TVAE | TabDDPM | TAEGAN |
| BW | CC | $0.995 \pm 0.002$ | $0.995 \pm 0.005$ | $0.997 \pm 0.003$ | $0.980 \pm 0.007$ | $0.932 \pm 0.004$ | $0.964 \pm 0.002$ |
| | DC | $0.989 \pm 0.004$ | $0.985 \pm 0.013$ | $0.942 \pm 0.015$ | $0.941 \pm 0.006$ | $0.945 \pm 0.023$ | $0.973 \pm 0.009$ |
| BT | CC | $0.990 \pm 0.004$ | $0.974 \pm 0.007$ | $0.964 \pm 0.007$ | $0.990 \pm 0.003$ | $0.946 \pm 0.004$ | $0.989 \pm 0.009$ |
| | DC | $0.975 \pm 0.014$ | $0.970 \pm 0.015$ | $0.923 \pm 0.015$ | $0.895 \pm 0.000$ | $0.956 \pm 0.023$ | $0.958 \pm 0.031$ |
| CR | CC | $0.998 \pm 0.002$ | $0.996 \pm 0.001$ | $0.974 \pm 0.004$ | $0.990 \pm 0.005$ | $0.963 \pm 0.001$ | $0.994 \pm 0.004$ |
| | DC | $0.995 \pm 0.003$ | $0.994 \pm 0.006$ | $0.980 \pm 0.003$ | $0.920 \pm 0.013$ | $0.958 \pm 0.001$ | $0.992 \pm 0.002$ |
| | DD | $0.988 \pm 0.003$ | $0.992 \pm 0.003$ | $0.969 \pm 0.002$ | $0.9338 \pm 0.003$ | $0.955 \pm 0.004$ | $0.973 \pm 0.003$ |
| DI | CC | $0.996 \pm 0.004$ | $0.993 \pm 0.003$ | $0.996 \pm 0.003$ | $0.985 \pm 0.003$ | $0.976 \pm 0.002$ | $0.987 \pm 0.007$ |
| | DC | $0.983 \pm 0.013$ | $0.980 \pm 0.007$ | $0.969 \pm 0.005$ | $0.977 \pm 0.005$ | $0.987 \pm 0.006$ | $0.954 \pm 0.004$ |
| IL | CC | $0.993 \pm 0.004$ | $0.993 \pm 0.005$ | $0.991 \pm 0.002$ | $0.978 \pm 0.008$ | $0.924 \pm 0.002$ | $0.996 \pm 0.005$ |
| | DC | $0.986 \pm 0.009$ | $0.993 \pm 0.005$ | $0.996 \pm 0.002$ | $0.991 \pm 0.006$ | $0.949 \pm 0.005$ | $0.986 \pm 0.012$ |
| | DD | $0.972 \pm 0.010$ | $0.980 \pm 0.001$ | $0.979 \pm 0.000$ | $0.898 \pm 0.058$ | $0.971 \pm 0.012$ | $0.977 \pm 0.014$ |
| QS | CC | $0.995 \pm 0.003$ | $0.980 \pm 0.002$ | $0.964 \pm 0.000$ | $0.969 \pm 0.002$ | $0.939 \pm 0.002$ | $0.997 \pm 0.001$ |
| | DC | $0.968 \pm 0.009$ | $0.986 \pm 0.004$ | $0.951 \pm 0.004$ | $0.907 \pm 0.005$ | $0.939 \pm 0.004$ | $0.985 \pm 0.004$ |
| SB | CC | $0.999 \pm 0.001$ | $0.995 \pm 0.000$ | $0.985 \pm 0.000$ | $0.956 \pm 0.002$ | $0.938 \pm 0.003$ | $0.985 \pm 0.001$ |
| | DC | $0.994 \pm 0.001$ | $0.993 \pm 0.003$ | $0.981 \pm 0.003$ | $0.899 \pm 0.004$ | $0.926 \pm 0.002$ | $0.960 \pm 0.003$ |
| WD | CC | $0.984 \pm 0.003$ | $0.941 \pm 0.000$ | $0.987 \pm 0.002$ | $0.950 \pm 0.007$ | $0.946 \pm 0.001$ | $0.997 \pm 0.001$ |
| | DC | $0.955 \pm 0.020$ | $0.935 \pm 0.007$ | $0.977 \pm 0.011$ | $0.938 \pm 0.009$ | $0.929 \pm 0.013$ | $0.997 \pm 0.003$ |
| AD | CC | $1.000 \pm 0.000$ | $0.994 \pm 0.000$ | $0.997 \pm 0.001$ | $0.985 \pm 0.001$ | $0.973 \pm 0.002$ | $0.994 \pm 0.000$ |
| | DC | $0.988 \pm 0.002$ | $0.978 \pm 0.001$ | $0.974 \pm 0.000$ | $0.947 \pm 0.001$ | $0.823 \pm 0.002$ | $0.966 \pm 0.000$ |
| | DD | $0.973 \pm 0.001$ | $0.981 \pm 0.001$ | $0.946 \pm 0.002$ | $0.970 \pm 0.001$ | $0.842 \pm 0.001$ | $0.949 \pm 0.000$ |
| CV | CC | $0.988 \pm 0.000$ | $0.999 \pm 0.000$ | $0.987 \pm 0.000$ | $0.979 \pm 0.001$ | $0.902 \pm 0.000$ | $0.992 \pm 0.000$ |
| | DC | $0.998 \pm 0.000$ | $1.000 \pm 0.000$ | $0.970 \pm 0.000$ | $0.983 \pm 0.001$ | $0.932 \pm 0.000$ | $0.989 \pm 0.001$ |
| | DD | $0.995 \pm 0.000$ | $1.000 \pm 0.000$ | $0.979 \pm 0.000$ | $0.989 \pm 0.000$ | $0.916 \pm 0.001$ | $0.992 \pm 0.000$ |

Table 8: Normalized correlation scores of each model. "D" stands for discrete, and "C" stands for continuous, so "DD" stands for the correlation between discrete columns, "DC" stands for the correlation between discrete and continuous columns, and "CC" stands for the correlation between continuous columns.

from transformers. GANs are of the next tier, still having almost all scores over 0.95, and TVAE and TabDDPM are slightly worse, yet still having all scores over 0.9.

| Dataset | Metric | non-deep-learning | | deep-learning | | | |
| --- | --- | --- | --- | --- | --- | --- | --- |
| | | ARF | REaLTabFormer | CTAB-GAN+ | TVAE | TabDDPM | TAEGAN |
| BW | Acc | **0.989 ± 0.008** | 0.977 ± 0.010 | **0.912 ± 0.034** | 0.878 ± 0.007 | 0.899 ± 0.024 | 0.869 ± 0.028 |
| | F1 | **0.993 ± 0.002** | 0.977 ± 0.021 | **0.913 ± 0.036** | 0.886 ± 0.007 | 0.893 ± 0.018 | 0.865 ± 0.029 |
| | Auc | **0.986 ± 0.014** | 0.981 ± 0.011 | **0.869 ± 0.021** | 0.840 ± 0.006 | 0.855 ± 0.015 | 0.826 ± 0.020 |
| BT | Acc | 0.907 ± 0.007 | **0.967 ± 0.003** | 0.851 ± 0.002 | 0.844 ± 0.014 | 0.831 ± 0.006 | **0.861 ± 0.016** |
| | F1 | 0.897 ± 0.013 | **0.963 ± 0.015** | 0.851 ± 0.002 | 0.847 ± 0.011 | 0.836 ± 0.006 | **0.860 ± 0.012** |
| | Auc | 0.883 ± 0.016 | **0.953 ± 0.019** | 0.821 ± 0.003 | 0.806 ± 0.005 | 0.817 ± 0.003 | **0.823 ± 0.006** |
| CR | Acc | 0.883 ± 0.003 | **0.990 ± 0.004** | 0.791 ± 0.005 | 0.775 ± 0.005 | 0.764 ± 0.002 | **0.895 ± 0.012** |
| | F1 | 0.869 ± 0.028 | **0.957 ± 0.010** | 0.781 ± 0.005 | 0.766 ± 0.005 | 0.754 ± 0.002 | **0.879 ± 0.010** |
| | Auc | 0.860 ± 0.020 | **0.987 ± 0.013** | 0.767 ± 0.002 | 0.759 ± 0.000 | 0.759 ± 0.000 | **0.846 ± 0.016** |
| DI | Acc | 0.916 ± 0.011 | **0.992 ± 0.008** | **0.934 ± 0.017** | 0.866 ± 0.018 | 0.901 ± 0.004 | 0.913 ± 0.010 |
| | F1 | 0.929 ± 0.010 | **0.970 ± 0.013** | **0.935 ± 0.017** | 0.874 ± 0.018 | 0.911 ± 0.004 | 0.917 ± 0.014 |
| | Auc | 0.909 ± 0.017 | **0.972 ± 0.007** | **0.921 ± 0.010** | 0.852 ± 0.012 | 0.822 ± 0.010 | 0.895 ± 0.002 |
| IL | Acc | 0.829 ± 0.025 | **0.896 ± 0.025** | 0.811 ± 0.009 | 0.807 ± 0.014 | 0.741 ± 0.003 | **0.819 ± 0.024** |
| | F1 | 0.845 ± 0.022 | **0.900 ± 0.018** | 0.831 ± 0.009 | 0.833 ± 0.012 | 0.768 ± 0.003 | **0.839 ± 0.023** |
| | Auc | 0.811 ± 0.015 | **0.883 ± 0.018** | 0.793 ± 0.014 | 0.794 ± 0.004 | 0.757 ± 0.000 | **0.798 ± 0.018** |
| QS | Acc | **0.786 ± 0.007** | 0.783 ± 0.002 | 0.748 ± 0.003 | 0.736 ± 0.002 | 0.734 ± 0.000 | **0.753 ± 0.007** |
| | F1 | **0.768 ± 0.008** | 0.767 ± 0.002 | 0.733 ± 0.003 | 0.720 ± 0.002 | 0.718 ± 0.000 | **0.737 ± 0.003** |
| | Auc | **0.782 ± 0.005** | 0.782 ± 0.002 | 0.768 ± 0.001 | 0.765 ± 0.000 | 0.765 ± 0.000 | **0.769 ± 0.003** |
| SB | Acc | 0.800 ± 0.007 | **0.876 ± 0.005** | 0.767 ± 0.004 | 0.756 ± 0.001 | 0.754 ± 0.000 | **0.775 ± 0.005** |
| | F1 | 0.799 ± 0.006 | **0.867 ± 0.002** | 0.767 ± 0.003 | 0.756 ± 0.001 | 0.754 ± 0.000 | **0.775 ± 0.004** |
| | Auc | 0.778 ± 0.002 | **0.837 ± 0.002** | 0.763 ± 0.001 | 0.761 ± 0.001 | 0.754 ± 0.000 | **0.765 ± 0.004** |
| WD | Acc | **0.878 ± 0.028** | 0.836 ± 0.003 | 0.826 ± 0.021 | **0.828 ± 0.006** | 0.756 ± 0.003 | 0.821 ± 0.016 |
| | F1 | **0.868 ± 0.017** | 0.844 ± 0.003 | 0.827 ± 0.009 | **0.833 ± 0.008** | 0.765 ± 0.003 | 0.827 ± 0.017 |
| | Auc | **0.858 ± 0.035** | 0.821 ± 0.000 | 0.804 ± 0.011 | **0.806 ± 0.007** | 0.770 ± 0.000 | 0.798 ± 0.016 |
| AD | Acc | 0.812 ± 0.000 | **0.945 ± 0.002** | 0.774 ± 0.001 | 0.794 ± 0.002 | 0.750 ± 0.000 | **0.857 ± 0.003** |
| | F1 | 0.807 ± 0.000 | **0.939 ± 0.002** | 0.771 ± 0.001 | 0.791 ± 0.002 | 0.747 ± 0.000 | **0.857 ± 0.003** |
| | Auc | 0.788 ± 0.001 | **0.928 ± 0.002** | 0.764 ± 0.000 | 0.766 ± 0.001 | 0.755 ± 0.000 | **0.816 ± 0.002** |
| CV | Acc | 0.783 ± 0.000 | **0.985 ± 0.003** | 0.768 ± 0.001 | 0.764 ± 0.001 | 0.750 ± 0.000 | **0.773 ± 0.001** |
| | F1 | 0.782 ± 0.000 | **0.973 ± 0.003** | 0.754 ± 0.000 | 0.753 ± 0.000 | 0.751 ± 0.000 | **0.774 ± 0.001** |
| | Auc | 0.760 ± 0.000 | **0.981 ± 0.003** | 0.754 ± 0.000 | 0.753 ± 0.000 | 0.751 ± 0.000 | **0.756 ± 0.000** |

Table 9: Normalized discrimination scores of each model. The best scores of all models, and of deep learning models are both in bold.

Table 9 gives the discrimination scores for the 10 datasets. LLM-based REaLTabFormer shows a clear advantage in this metric, especially on large datasets. ARF also performs quite good, which is also expected because ARF is tuned based on a random-forest discriminator, which is expected to perform better than MLP discriminators in GANs. Among all deep learning models, GANs perform better in this metric, and TAEGAN performs generally better than CTAB-GAN+. In other words, the conclusion made in Section 4.3 still holds for discrimination metric.

