# OpenReview forum: "TAEGAN: Generating Synthetic Tabular Data for Data Augmentation"
_ICLR.cc/2025/Conference — Submitted to ICLR 2025_

### Official Review · Reviewer_GLTJ · 2024-11-01

**Soundness:** 2
**Presentation:** 1
**Contribution:** 2
**Rating:** 3
**Confidence:** 4

**Summary:**

The paper proposes TAEGAN, a new GAN-based method for tabular data generation. In this work, the authors focus on the problem of generating tabular data for the sake of data augmentation, which differentiates this paper from the majority of the other papers in the field as they focus on the problem of using the synthetic datasets to *replace* the original one. The novelty of TAEGAN lies in the generator model, which is replaced by a masked auto encoder. The method is compared against standard state-of-the-art models including CTAB-GAN+, TVAE, TabDDDPM etc etc and manages to outperform them in most scenarios.

**Strengths:**

The paper poses an interesting problem and the method manages to outperform some state-of-the-art model.

**Weaknesses:**

First of all, the paper needs some rewriting. Indeed, it is written in a very colloquial way and without the Figures and the Examples in would be very difficult to understand wha the authors mean. For example, at page 3 the authors introduce the mode-specific normalisation introduced in Xu et al. (2019). While the text gives an intuition of what that means, it does not define the process and only a reader that knows about it would be able to follow. Also, at times the flow of the paper is interrupted or difficult to follow. For example, in page 4 the authors talk about CTAB-GAN+ and how in the mask vector *m* only one dimension can be 1. It was not clear to me why it this specified there until I reached page 5 where the authors explain that in TAEGAN this assumption is dropped.

Secondly, there are certain parts of the experimental analysis that are not convincing:
- First of all, why did the authors compare only with TVAE and not also with CTGAN? I am asking about this model in particular because it is available in the same repo as TVAE and in many cases performs better than TVAE.
- Secondly, if the focus of the paper is to do data augmentation for very small datasets, why did you not compare with GOGGLE? It is a model that is supposed to perform particularly well on smaller datasets
- Thirdly, in the evaluation only a machine learning model is used (namely XGBoost). This is a bit weird for me, as normally a suite of 3 or 5 models are used (including XGBoost, Random Forest, Decision Trees etc etc)
- Then, another common metric used to evaluate this models is the sampling generation time. Could you specify how your architecture perform in this respect?
- In appendix B.3.1 the authors mention 4:1 split. Could the authors clarify what they mean here and why they used this specific split?
- Finally, the difference in performance even in the data augmentation case is very small. Could the authors provide some analysis of the statical significance of their results? (e.g., using the wilcoxon test)

**Questions:**

See above.

---

### Official Review · Reviewer_whjk · 2024-11-02

**Soundness:** 3
**Presentation:** 3
**Contribution:** 3
**Rating:** 8
**Confidence:** 4

**Summary:**

This paper proposes a new MAE and GAN based method for generating synthetic tabular data. In particular, the authors focus on the use of synthetic data used for augmenting small datasets. The main contribution is the architecture and sampling process of generator model. The authors make a number of modifications to previously used GAN-based architectures, such as using an AE instead and modifying the masking mechanism to be multivariate instead of a single feature. While the "quality" of the synthetic data itself is not as strong as that of a transformer-based architecture, the experimental results show that the proposed model is indeed strong in augmentation in low-data scenarios.

**Strengths:**

- The paper is organized well and is easy to follow, with detailed explanations on the architecture (including the sampling and training procedures) and evaluation methods.
- The use case of augmenting small datasets is a good choice and addresses a gap in current literature.
- The proposed architecture introduces a novel combination of a masked auto-encoder with GANs for generating tabular data, which is a creative departure from standard GANs and LLM-based approaches.

**Weaknesses:**

- While the authors acknowledge that some parts of the architecture does not scale well for larger (both row and column-wise), the experiments do not consider this. Further investigation into how the different methods scale (runtime, compute etc.) would be nice.
- While the model performs well on classification metrics, a deeper analysis into the generated data’s quality (e.g., assessing diversity or fidelity independently) might provide a clearer picture of its augmentation effectiveness. For instance, additional metrics could be explored to measure how close the generated samples are to the original data distribution, or additional analysis as to how TAEGAN generated data differs from different approaches could reveal interesting insights.
- The paper mentions various parameters, such as the interaction loss weight, reconstruction loss scaling, and mask sampling parameters. More details into the HPO process would be nice.

**Questions:**

- While interesting, the mask sampling strategy adds complexity. Could the authors discuss how this affects training time and whether there are scenarios where a simpler sampling method could suffice without significantly compromising model quality?
- Similarly, can impact of interaction loss be discussed in terms of the training performance?

---

### Official Review · Reviewer_N6vm · 2024-11-02

**Soundness:** 2
**Presentation:** 2
**Contribution:** 2
**Rating:** 5
**Confidence:** 3

**Summary:**

The paper proposes TAEGAN for generating synthetic tabular data. The key difference to prior work is to introduce a masked auto-encoder as the generator, hoping to bring the benefits of self-supervised training into the GAN training process. The paper evaluates the results on 10 datasets and demonstrates the state-of-the-art performance on data augmentation tasks. TAEGAN is also better than non-LLM approaches in train-on-synthetic-test-on-real evaluation.

**Strengths:**

* The paper evaluates the results on a wide range of datasets and demonstrates clear benefits over non-LLM approaches.

* Some figures in the paper are nicely drawn, explaining the complicated processes clearly.

**Weaknesses:**

* The claim that LLMs are more prone to overfitting and are not suitable for data augmentation applications is questionable.

* The approach introduces several new hyper-parameters, and more hyper-parameter studies are needed.

* The motivation of some designs is not clearly explained.

Please see the next section for more details about these questions.

**Questions:**

* The paper claims that "LLM-based models are expected to produce data with very high fidelity, but the way it is trained, which is by na¨ıve next-token-predictions, makes it hard to extend the generator’s ability beyond the training set" and "In comparison, previous state-of-the-art models in the field, generative adversarial networks (GANs) (Goodfellow et al., 2014), can excel in generalization" is not rigorous enough. GANs can also memorize training samples (https://arxiv.org/abs/2210.12231) and LLMs can also generalize well (https://arxiv.org/pdf/2305.13673). Since there are many variables, we need to state the setting clearly when making such claims. Especially, model sizes play an important role in memorization---we can make LLM small enough so as to reduce their memorization issues.

* That being said, the conclusion in the experimental section that "The fact that TAEGAN (or GANs in general) performs particularly well for data augmentation on small datasets suggests that, while GANs may not capture internal data relationships as effectively as LLMs, they avoid being overskill, with model sizes only around 2-5% of LLM." and "LLM-based REaLTabFormer tends to be better" on data quality is valid, but not that interesting enough. As discussed in the paper, these results could simply be due to the artifact of the model sizes, instead of the fundamental difference in the properties of the two algorithms. A more informative comparison would be to vary the sizes of both models show how the data augmentation (4.2) and data quality (4.3) metrics evolve with model sizes, and compare the trade-off between data augmentation, data quality, and model size. For example, if we modify the parameters of REaLTabFormer so that it has the same size as TAEGAN in the experiments, whether REaLTabFormer could also excel in data augmentation tasks?

* I like Figure 2 and Figure 3 very much, as they explain "how" the approach works in a very clear way. However, the paper lacks discussions about "why" they are designed this way for some of the proposed methods. For example,
    * The first paragraph of Section 3.3.1 explains that multiple components can be selected. Why is it beneficial?
    * The last paragraph of Section 3.3.1 and Figure 3b explains how the weights of the rows are computed. But why are they designed this way?

* The proposed approach introduces some new hyper-parameters, including \tau in Section 3.3.3 and the varying temperature in Section 3.4. Do you use the same value across all experiments and how do you choose the values? How sensitive is the algorithm concerning these hyper-parameters?

* In addition, how do you choose the hyperparameter of XGBoost in the experiments? Do you use the same hyperparameter across all experiments?

* The ablation study does not cover all important components of the algorithm. For example, what would the performance be if we removed the masked autoencoder while keeping all other components?

* The second to the last paragraph in Section 3.3: \mu_i and \mu_j are defined as the expanded mask in Section 3.1. It contains no generated data values. I don't understand why applying losses on them is useful?

---

### Official Review · Reviewer_kE8E · 2024-11-03

**Soundness:** 2
**Presentation:** 1
**Contribution:** 2
**Rating:** 3
**Confidence:** 2

**Summary:**

This paper propses a novel GAN architecture for tabular model generation. An enhanced multivariate sampling method on tabular data is propsed to address data skewness in all features instead of generating date base on only one feature at each time. However, the experimental results show that the improvements achieved by the proposed method is quite limited.

**Strengths:**

The multivariate sampling method is novel and shows slight improvement compared to random sampling.

**Weaknesses:**

* There exists many details that are left unclarified, making it difficult for readers to comprehend the intuition or recreate the experiments,
* The authors claimed that TAEGAN is better in augmentation tasks, but LLMs still performed baset in the realted tasks.
* The porposed algorithm is only suitable for smaller datasets, as it takes O(Ｎ^2) time when the given dataset has N features.

**Questions:**

* During the Inferencing sampling process, why do yiou unmask each feature one by one in a loop instead of feeding the entire hint into the generator to generate a sample at once?
* Interaction Loss was introduced, but never explained in detail. I don't see where this loss was used in TAEGAN.
* The authors dIdn’t explain how the weights in weight vector of continuous components were calculated.
* How do you handle the common problems of GAN training, like mode collapse and model convergence. More experimental results are necessary to justify diverserity.
* Graph Flow is unclear in FIG.2 The arrow from the original dataset links to the hint vector training table, this is misleading because the training table was constructed by sampling dataset M times. The arrow will make readers think that there exists a one to one relation between the rows.

---

### Official Review · Reviewer_jjqZ · 2024-11-04

**Soundness:** 2
**Presentation:** 1
**Contribution:** 1
**Rating:** 3
**Confidence:** 4

**Summary:**

The paper introduces TAEGAN, a GAN-based framework for generating synthetic tabular data, specifically designed for data augmentation in small datasets. TAEGAN uses a masked auto-encoder as its generator, enabling it to benefit from self-supervised pre-training to improve data generation quality. Unlike large language models (LLMs), which often require extensive computational resources and are less effective for small datasets, TAEGAN is optimized for smaller data settings, aiming to improve machine learning model performance with augmented data.

**Strengths:**

One strength of this paper is the range of analyses conducted on the proposed model, including various types of ablation studies. These analyses provide valuable insights into the model’s functionality. However, it would be beneficial to include more detailed analyses to further understand the effects of individual components and hyperparameters on the model’s performance.

**Weaknesses:**

1. $\textbf{Representation}$: The paper allocates excessive space to explaining existing approaches, such as CTGAN and CTAB-GAN+, rather than focusing on how TAEGAN differs from these methods and the specific benefits gained from its unique components. Additionally, instead of detailed textual descriptions for elements like the loss function and the proposed method, a clearer presentation through mathematical formulas would improve comprehension and conciseness.

2. $\textbf{Novelty}$: TAEGAN’s approach closely resembles existing work in the field, limiting the novelty of its contribution. The modifications made, while functional, may not be significant enough to distinguish it from prior GAN-based methods for tabular data generation.

3. $\textbf{Loss of Information in Continuous Variable Binning}$: The relaxation technique used in TAEGAN, which involves binning continuous variables, may lead to considerable information loss. This could affect the accuracy and quality of generated data, particularly for datasets with nuanced or fine-grained continuous features.

4. $\textbf{Limited Performance Improvement over Baselines}$: TAEGAN shows only marginal performance gains over baseline models in both primary experiments.

**Questions:**

1. What is the parameter search space for the proposed method? It would be helpful to understand whether a similar level of parameter tuning was applied to the baseline models as well. Additionally, the performance differences between baselines in Table 3 appear to diverge from the community’s understanding of these models. Specifically, diffusion models are generally known to outperform GAN- and VAE-based models like TVAE. Could you clarify these discrepancies?

2. It would be thankful if the authors can provide oversampling performance as well.

---

### Official Review · Reviewer_cTTz · 2024-11-04

**Soundness:** 2
**Presentation:** 2
**Contribution:** 2
**Rating:** 5
**Confidence:** 2

**Summary:**

The paper presents TAEGAN, a GAN-based framework for generating synthetic tabular data for effective data augmentation. It introduces a masked auto-encoder generator, leveraging self-supervised pretraining to improve the model’s understanding of data relationships. TAEGAN is optimized for small datasets, where traditional large models are often less effective, and demonstrates superior performance in both data quality and augmentation on smaller datasets compared to several state-of-the-art models. Experimental results show TAEGAN’s efficacy, particularly in scenarios with limited data, supporting various ML tasks.

**Strengths:**

- Demonstrates effective data augmentation capabilities, particularly for small datasets, outperforming other models on most tested datasets.
- Incorporates a masked auto-encoder generator, which improves inter-feature relationship learning, enhancing data quality.
- Introduces multivariate training-by-sampling, addressing data skewness effectively across multiple feature types.
- Provides thorough empirical evaluations and ablation studies, showing the robustness of design choices.

**Weaknesses:**

- In Table 2 and Table 3, TAEGAN’s performance is very close to the best baseline (typically within 1% or comparable), raising questions about the significance of the proposed improvements.
- In Table 3, the performance of TAEGAN on larger datasets, particularly the AD and CV datasets, is worse than the LLM-based REaLTabFormer method, which indicates TAEGAN's limitations in scalability for larger datasets.
- Efficiency comparison between TAEGAN and other baselines are not discussed. The sequential component generation in sampling and the interaction loss calculation in TAEGAN may be computationally intensive.
- The approach remains GAN-dependent, which may feel outdated given the current advances in LLMs, and poses challenges like mode collapse.

**Questions:**

- In Table 3, it is somewhat confusing to categorize REaLTabFormer as a non-deep-learning method, given that it is an LLM-based method. Clarification on this point would be helpful.
- How effective TAEGAN is compared to the simple LLM-based synthetic data generation baseline (without any LLM training)? For example, prompt engineering (e.g., few-shot demonstrations for a specific dataset generation task) could leverage state-of-the-art LLMs (like GPT-4 or open-source models) to generate samples, especially in small dataset scenarios where a few-shot demonstration does not require a large sample set.

---

### Meta-Review · Area_Chair_3hDt · 2024-12-16

**Metareview:**

This paper proposed a GAN-based tabular data synthesis method after observing recent LLM-based tabular data synthesis methods are too large when considering their target tabular data size small. This paper's GAN design is outdated (as pointed by some reviewers) and they did not conduct enough experiments with recent methods. They only compared with TabDDPM but there are more recent diffusion-based methods that are not as large as LLMs. They also conducted experiments with a small number of datasets. They need to consider more baselines and datasets. Before that, their main motivation, i.e., recent models are not suitable for small or sparse tables, does not make sense with enough justifications. In order to support this motivation, they need to show appropriate literature survey and statistics. Reviewers also raised issues on the effectiveness and the timeliness of the proposed method since the method is based on an outdated GAN-based framework and their claim that LLMs are prone to overfitting is not properly justified.

**Additional Comments On Reviewer Discussion:**

The authors did not participate in the rebuttal discussion.

---

### Decision · Program_Chairs · 2025-01-22

Reject